# OpenReview forum: "On the Identifiability of Concepts from Large Language Model Activations"
_ICLR.cc/2026/Conference — Submitted to ICLR 2026_

### Official Review · Reviewer_YXnW · 2025-10-30

**Soundness:** 3
**Presentation:** 4
**Contribution:** 4
**Rating:** 8
**Confidence:** 3

**Summary:**

The paper tackles non-identifiability in SAEs, introducing Sparse Shift Autoencoders (SSAE), which learn interpretable, steerable directions by training on differences between two LLM activations rather than on single states. Under strong but plausible assumptions, the authors argue these directions are identifiable (up to permutation/scale) and can be injected to nudge generation toward desired, human-meaningful behaviours without finetuning.

**Strengths:**

Identifiability (up to perm/scale) ties a math guarantee to usable vectors. Even if under strong assumptions, it nudges the field toward principled representation editing rather than ad-hoc hacks.

Out-of-distribution wins suggest the method isn’t just memorising and hints, the learned axes reflect model-internal structure.

Works with weakly paired data instead of per-concept labels, lowering supervision cost for discovering useful steering axes.

**Weaknesses:**

Pairs are constructed to alter a target concept, and supervision is felt to have shifted to the data generation process.

Identifiability leans on linearity and coverage that likely hold only near the last layer and for neat attributes.

**Questions:**

N/A

---

> ### Author Response · Authors · 2025-11-21
> **Response to Reviewer YXnW**
>
> We thank you for your review of our work; we are delighted that you find our method as “nudging the field toward principled representation editing rather than ad-hoc hacks”, consider the "out-of-distribution wins” as indicating that the model doesn’t just memorise, and that the method is based on “lowering supervision cost”. Below, we we engage with the questions and nuanced points raised in your review, and we welcome further discussion on these aspects:
>
> > _Pairs are constructed to alter a target concept, and supervision is felt to have shifted to the data generation process._
>
> We clarify the construction of the pairs: these pairs don’t need to differ in just the target concept and our real-world data experiments are conducted by randomly pairing samples from existing datasets without any rejection sampling.
>
> > _Identifiability leans on linearity and coverage that likely hold only near the last layer and for neat attributes._
>
> This is accurate and is due to the fact that the linear representation hypothesis hasn’t been proven yet for the intermediate layers of an LLM. Empirically, it’s been observed that concepts are linearly separable in intermediate layers, and SAEs are usually trained on middle layers’ activations. We provide a study of recovering concepts across different layers of an LLM in Appendix B.5 to test the robustness of our method to Assumption 1 on the Linear Representation Hypothesis not necessarily being satisfied.

---

> ### Author Response · Authors · 2025-11-27
> **Request for Response**
>
> Dear Reviewer YXnW, just a gentle reminder to please take a moment to respond to our rebuttal when you have the chance. Your engagement will help us improve our work! Thank you again for your time and feedback.

---

### Official Review · Reviewer_AEzo · 2025-11-01

**Soundness:** 3
**Presentation:** 3
**Contribution:** 2
**Rating:** 4
**Confidence:** 3

**Summary:**

The paper introduces a variant of the Sparse Autoencoder (SAE) architecture, called Sparse Shift Autoencoders (SSAE). Architecturally, the SSAE differs from (Cunningham et al, 2023)-style SAEs in three ways: 1. the SSAE autoencodes *differences* of hidden states, instead of the hidden states themselves, 2. the SSAE is entirely affine, instead of having an activation function, and 3. the SSAE is trained with an explicit l_1 norm constraint on the hidden layer, instead of using the l_1 norm in the loss term. These changes are made in an attempt to improve the identifiability of the learned features, which the authors primarily measure by the Mean Correlation Coefficient (MCC). The paper has both theoretical (proof-based) and empirical results. The theoretical results show that if concepts are represented linearly, an SSAE with perfect reconstructions and minimal l_1-norm essentially consists of identifying the transformation from representation space to concept space. The empirical results show that the discovered concepts can be used to steer, and that they have high MCC with "ground truth" features.

**Strengths:**

The paper contains both theoretical and empirical work. The theoretical work (Propositions 1 and 2) are intriguing results about the optimal reconstructions that would be found by a linear autoencoder. The authors do an excellent job explaining their formal assumptions, and justifying why they should be granted.

The authors compare their SSAEs to a robust set of baselines, including pre-trained SAEs, and naive approaches like PCA and Mean Differences.

**Weaknesses:**

The paper's theoretical section is based on assuming perfect reconstruction accuracy, which SAEs do not achieve in practice, and the authors do not address how error terms would propagate through the argument. This makes it hard to assess how this argument would apply to real-world models, where reconstruction accuracy is always imperfect.

The SSAE is trained on semi-labelled data, where data is pre-sorted to be identical words except with changed gender and/or language (e.g. ("Taureau","Cow") in the BINARY(2,2) dataset).

All SSAE experiments in the body of the paper happen on datasets one or two varying concepts. Thus (assuming the linear representation hypothesis holds), the SSAE is given a relatively easy task of learning the vectors x and y from a dataset of the form {x,y,x+y}. The paper does include an appendix B.7 in which SSAEs are tested on synthetic datasets with up to 10 concepts, but they are not baselined against SAEs. However, prior work such as (Sharkey et al, 2022) showed that SAEs are able to identify ground-truth directions even on a more complicated synthetic dataset, with a comparable metric of Mean Maximum Cosine Similarity ≈ 1 with the correct choice of hyperparameters.

The authors evaluate SSAEs for steering on two metrics: steering strength (Figure 2), and Mean Correlation Coefficient (MCC) (Figures 3-5). However, both of these metrics have flaws:

- Steering is done by adding a scalar multiple of a vector to the activations, but the authors appear to be comparing unnormalized vectors in Figure 2, so the quantity of "steering strength" is not comparable between vectors. In other words, if the features used for steering in Figure 2 are v and w for the SSAE and SAE respectively, then steering with the feature 2w would make the SAE's performance appear nearly identical to the SSAE's. Prior work has evaluated steering by the extent of steering possible without harming downstream performance, for instance in (Durmus et al, 2024) for SAE feature steering, and (Lamb et al, 2025) Table 1 for non-SAE feature steering.

- The authors never establish that MCC is a useful metric for measuring the quality of directions found. Indeed, in Appendix B.4 the authors note "that MCC is insufficient as a standalone criterion for model comparison: it cannot distinguish between representations that differ in their capacity to support reliable steering".

**Questions:**

**Questions**

1. Line 138 and Line 1185: Is Assumption 1 the same as Assumption 5? If so, would it be possible to remove Assumption 5 so that references like the one in line 217 do not require reading ahead?

2. Line 995:  The paper says "since \hat r and \hat q are linear..." but the hypotheses of the proposition only assume \hat r and \hat q are affine, not linear. Is there some other reason the bias terms on \hat r and \hat q should be 0, or does the statement need to be tweaked to allow bias terms?

3. Do Propositions 1 and 2 ever make use of the fact that \delta^z are shifts? Or would these make results apply to any linear SAE under Assumptions 1-3?

4. Your proof of Proposition 2 makes use of the fact that zero reconstruction loss is attainable, so all solutions must satisfy \hat q(\hat r(z))=z. Do you think this result is robust to errors? In particular, if the reconstruction loss is small in expectation, can you give a bound on how far \hat q and \hat r will be from the described shapes?

5. Relatedly, in your empirical tests, what reconstruction error (or what Fraction of Variance Explained) do your SSAEs attain?

6. In Line 322, when computing MCC, what is the "true" latent dimension you measured against?

----

**Small Comments**

The following changes would make small improvements to the paper:

Line 218: Please include a comment in the body of the paper that A_V+ denotes the pseudo-inverse. While you do describe your notation in an appendix, there is no indication that the reader should look there.

Line 1400: "Figure 10 and ??" seems to be missing a reference.

Line 1403: The sentence "On TruthQA, the MCQ track" is incomplete.

Lines 1520 and 1524: There seem to be missing close-parentheses in "abs(corr(x1,y1) + abs(corr(x2,y2)" and "mean(abs(corr(x1,y1),abs(corr(x2,y2))".

Line 286: (Biderman et al 2023) do not study SAEs. Was this intended to be (EleutherAI, 2023)?

----

**Additional References**

(Durmus et al, 2024) Evaluating feature steering: A case study in mitigating social biases. https://www.anthropic.com/research/evaluating-feature-steering

(Lamb et al, 2025) Focus On This, Not That! Steering LLMs with Adaptive Feature Specification. https://openreview.net/forum?id=rbI5mOUA8Z

(Sharkey et al, 2022) [Interim research report] Taking features out of superposition with sparse autoencoders https://www.alignmentforum.org/posts/z6QQJbtpkEAX3Aojj/interim-research-report-taking-features-out-of-superposition

---

> ### Author Response · Authors · 2025-11-21
> **Response to Reviewer AEzo**
>
> We thank you for your review of our work; we are delighted that you acknowledge the architectural novelty of our work, recognise both its theoretical and empirical aspects, and consider we did “an excellent job explaining formal assumptions” and justifying them. Below, we we engage with the questions and nuanced points raised your review, and we welcome further discussion on these aspects:
>
> > _The paper's theoretical section is based on assuming perfect reconstruction accuracy, which SAEs do not achieve in practice, and the authors do not address how error terms would propagate through the argument. This makes it hard to assess how this argument would apply to real-world models, where reconstruction accuracy is always imperfect._
>
> Across all SSAE models (our method) trained in the paper, the reconstruction error is effectively zero ($< 1e-3$). We will include an appendix table reporting the exact reconstruction-error values across all datasets and checkpoints for clarity.  By contrast, it is correct that SAEs do not generally achieve zero reconstruction error in practice. This is primarily because their training objective balances reconstruction with a sparsity regularisation term, making optimisation non-trivial and often leading to underfitting (see, e.g., [1]).
>
> Importantly, the theoretical argument in Section 4 assumes perfect reconstruction since identifiability guarantees are stated in the limit of infinite data; relaxing this assumption introduces additive error terms that do not qualitatively change the identifiability conclusion.
>
> > _The SSAE is trained on semi-labelled data, where data is pre-sorted to be identical words except with changed gender and/or language (e.g. ("Taureau","Cow") in the BINARY(2,2) dataset)._
>
> The SSAE is also trained on multiple real-world datasets: TruthfulQA, Sycophancy, Refusal, and BiasinBios (see Figs 2 and 5 for reference).
>
> > _All SSAE experiments in the body of the paper happen on datasets one or two varying concepts._
>
> Most real-world datasets considered in the interpretability community for the recovery of steering vectors consist of a single concept (for eg TruthfulQA, Sycophancy, Refusal) [2, 3] thus we evaluate on these datasets. We will add results comparing SSAEs and SAEs on the synthetic datasets in the camera ready version of the paper in Appendix B.7.
>
> > _The authors evaluate SSAEs for steering on two metrics: steering strength (Figure 2), and Mean Correlation Coefficient (MCC) (Figures 3-5). However, both of these metrics have flaws:_
>
> We compare steering strengths considering normalised vectors to compare across models.
>
> Re: MCC, we agree it is not sufficient as a standalone evaluation metric for downstream steering quality, as we explicitly state in Appendix B.4. Our use of MCC is for a different purpose: assessing identifiability of the learned decoder, particularly in settings without ground-truth concept labels. MCC is the standard metric for this purpose in the identifiable representation learning literature, because an MCC score close to 1 indicates that the recovered steering vectors match the ground-truth directions up to permutation and scaling. To avoid confusion, we also supplement MCC with downstream behavioural evaluations. As shown in Table 1 and Figure 3, higher MCC values consistently correspond to greater cosine similarity between the target activation shift and the shift induced by steering with the decoder-derived direction. Thus, while MCC is not intended to measure full steering efficacy, it is an appropriate and widely used measure of identifiability and unsupervised model selection, and our empirical results confirm that higher identifiability (as measured by MCC) aligns with more accurate steering directions.
>
> _(Response continued in further comments due to character limit)_
>
>
> [1] [Position: Adopt Constraints Over Penalties in Deep Learning; Ramirez, et al](https://arxiv.org/abs/2505.20628)
>
> [2] [Steering Llama 2 via Contrastive Activation Addition; Panickssery, et al](https://arxiv.org/abs/2312.06681)
>
> [3] [TruthfulQA: Measuring How Models Mimic Human Falsehoods; Lin, et al](https://arxiv.org/abs/2109.07958)

---

> ### Author Response · Authors · 2025-11-21
> **continued response**
>
> > _Questions:_
>
> - Thank you for your suggestion, we will fix the typo in the Assumption naming, they are indeed the same assumption.
>
> - Thanks for your question! For an identifiability proof (as you can see in Appendix A.5 as well), there is no difference between linear and affine.
>
> - $\delta^z$ are simply vectors such that they satisfy equation (5) for any pair of concept representations that follow the data generating process in equations (2) and (3). In practice, this amounts to pairing any two input activations to an SAE for a sufficiently large number of samples.
>
> - Non-parametric identifiability doesn’t work with bounds. There are no finite sample guarantees in theory. If you read identifiability literature [4, 5, 6], all proofs provide exact guarantees.
>
> - As we have noted earlier, all considered models in the paper achieve nearly zero reconstruction error (~1e-3). We will add a table with the exact numbers in the appendix, thank you for your suggestion.
>
> - We use the true latent dimension as the number of concepts a simple semi-synthetic dataset is designed to contain. For eg: _Gender(1, 1)_ has 1 concept of gender varying across all samples. For real world datasets, we can consider the encoding dimension to be equal to the true latent dimension, such as 1 for sycophancy, refusal, TruthfulQA, since in each of these datasets there is only one concept of interest. In the paper, we present experimental results both for this case and when the latent dimension is greater than 1, for eg: see the paragraph on “Real world steering”.
>
> > _Small comments:_
>
> Thank you for your thorough reading of our paper, we will make these improvements to improve the readability of the paper. Re: the citation for Pythia, the [[Biderman 2023]](https://proceedings.mlr.press/v202/biderman23a/biderman23a.pdf)  paper points to “Pythia: A Suite for Analyzing Large Language Models Across Training and Scaling”. Can you please share what the EleutherAI, 2023 reference is?
>
> [4] [Disentanglement via Mechanism Sparsity Regularization: A New Principle for Nonlinear ICA; Lachapelle, et al](https://proceedings.mlr.press/v177/lachapelle22a.html)
>
> [5] [Weakly Supervised Representation Learning with Sparse Perturbations; Ahuja, et al.](https://arxiv.org/abs/2206.01101)
>
> [6] [Variational Autoencoders and Nonlinear ICA: A Unifying Framework; Khemakhem, et al.](https://arxiv.org/abs/1907.04809)

---

> ### Author Response · Authors · 2025-11-27
> **Request for Response**
>
> Dear Reviewer AEzo, just a gentle reminder to please take a moment to respond to our rebuttal when you have the chance. Your engagement will help us improve our work! Thank you again for your time and feedback.

---

### Official Review · Reviewer_whSt · 2025-11-02

**Soundness:** 2
**Presentation:** 1
**Contribution:** 1
**Rating:** 2
**Confidence:** 3

**Summary:**

The authors propose a new kind of Sparse Autoencoder called SSAE that they train on activation differences between to prompts with known, different concepts. Example concepts: language, gender, truth. They hope that SSAE produces better steering vectors than ordinary SAEs.

The paper suffers from bad presentation and a lack of empirical evidence, especially showcasing what their model learned and comparing it to existing approaches. I have some concerns and questions regarding the method that I'd like to rebut.

**Strengths:**

- it is well-known that SAE features aren't great steering vectors and the authors tried to tackle this important problem

**Weaknesses:**

- The authors frame SSAE as an improvement of ordinary SAE. Yet, they don't show basic data about their models. For example, I'd like to see reconstruction error (variance unexplained), L0, feature dashboards, auto-interp, etc, and a side-by-side comparison to SAEs.
- When presenting a new SAE approach, I'd like to see at least one of two things: (1) the new SAE is better. Then, this needs to be proven empirically, or (2) the new SAE can answer scientific questions we are interested in that normal SAEs could not. This would also need to be shown empirically. I didn't see evidence for either.
- Training data need to be pairs where samples have known concepts. This is an important difference to SAEs which you can train unsupervised from activations alone. This limits the applicability of the method quite a lot because we don't have datasets for most concepts.
- The author's goal is to steer and they use a supervised dataset with labelled concepts to steer (which SSAE require). However, there are also other methods that can be used to find steering vectors in this setting, for example distributed alignment search. The authors don't compare their SSAE to steering methods that utilize the same dataset.
- I have some more concerns about the methodology but I put it under "Questions" below.


I found the presentation and the writing quite poor throughout the manuscript. Here are some examples to give the authors a better impression of why I found the paper very difficult to read:
- the title is generic and says nothing. What is the one result or idea that you'd like to convey?
- imprecision and missing definitions. ll 016 "SAEs are not guaranteed to be identifiable". What does "identifiable" mean? Do you really mean SAEs as a whole or rather SAE features? Why is an absolute guarantee important in the messy world of LLMs?
- the paper could be cut in size, e.g. sentences like "In this paper, we bring the question of identifiability to the forefront of LLM interpretability research" could be cut and replaced by e.g. a sentence explaining what identifiability is and why it's a big problem
- some words raise confusion, for example you say "token embeddings" which typically means the word embedding vectors before layer 0. Say "activation vector" or "representation of residual stream" for better understanding.
- important cites are missing, for example Bricken 2023 introduced SAEs independently and around the same time as Cunningham.
- math is confusing and used heavily throughout the manuscript. However, in most places, it's not needed and just makes the paper very hard to read.
- important information that would guide understanding like what the dataset is, at least what kind of data it contains, is only given briefly at the end of the paper. Showing data and how it's processed is often easier for the reader's understanding than frontloading unnecessary math.

**Questions:**

I have quite a lot of questions that I couldn't figure out yet by reading the paper alone.
- What is "identifiable"?
- In almost all cases, the authors train on only one single concept. Why would we need to create the entire SSAE if we only have one concept? Isn't the whole point of SAEs that we can learn millions of concepts at the same time? Why can we not use methods made for this use case like distributed alignment search (DAS)?
- Different to vanilla SAEs, there's no nonlinearity (i.e. ReLU) that could guide sparsity, it's only two linear maps. Why was no ReLU used? How can sparsity develop without ReLU? Activations point into different directions and have different scale. If you feed them through a linear map, they should still point into different directions at different scales (unless you defflate them all to zero in which case you couldn't recover anything). But then, it's virtually impossible for most values to be zero (i.e. sparse). Without ReLU, you might have low activations (because of l1 penalty) but they are still dense, not sparse. Can the authors report L0 metric and clarify how their architecture can learn a sparse code?
- Why should the learned representation have size V (ll179)? What do you mean by "learned representation" here? Features? Reconstructions? I'd expect there to be many more concepts than the activation space has dimensions because of superposition.
- Why were activations only taken from the last token in context (ll305)? Unlike e.g. BERT, decoder-only LLMs don't produce a sentence representation at their last token.
- ll322 what is "true latent dimension"?

---

> ### Author Response · Authors · 2025-11-21
> **Response to Reviewer whSt**
>
> We believe that you have misunderstood fundamental aspects of our work. As the title indicates and as we define multiple times in the paper, this is first and foremost a novel theoretical and algorithmic contribution developing conditions under which sparsity-regularized auto-encoding can provably find latent concepts from LLM activations in an unsupervised way. We note that other reviewers clearly recognize this contribution and its novelty, stating that the paper is “solidly backed up by conceptual and theoretical arguments” (XVbS) with “an excellent job explaining formal assumptions” (AEzo), shows experiments backing the claims (XVbS, AEzo, YXnW) with “out-of-distribution wins” (YXnW), and contributes to the principled “improvements to SAEs” (XVbS, YXnW) with its architectural novelty (AEzo). Below, we address your questions and concerns in an attempt to engage in constructive dialogue:
>
> > _The authors frame SSAE as an improvement of ordinary SAE. Yet, they don't show basic data about their models. For example, I'd like to see reconstruction error (variance unexplained), L0, feature dashboards, auto-interp, etc, and a side-by-side comparison to SAEs._
>
> The expectation of detailed dashboard-style empirical evaluation for a paper whose primary claims are theoretical seem to arise from a mismatch between what the paper aims to contribute and the evaluation paradigm you have in mind. Our work is not an empirical benchmark, but a phenomenological and theoretical contribution: it proposes an identifiable model and a practically realisable architecture for interpreting concepts from LLM activations. The goal of the paper is not to argue that SSAEs outperform SAEs on downstream tasks, but to formalise when and why identifiability of concepts is possible, and how.
>
> We did not report reconstruction error because for all model checkpoints presented in the paper, it is effectively zero. However, we will upload a revised paper that includes an appendix table reporting the exact values across all datasets for full transparency.
>
> Additionally, evaluation metrics such as reconstruction error, $\ell_0$, feature dashboards, qualitative auto-interp, etc. are _not_ meaningful indicators of identifiability. Indeed, the key observation made by long-standing identifiability literature is that  _a zero reconstruction error does not imply that the learned features correspond to the correct underlying concepts_, motivating the extra constraints we propose. In fact, even unconstrained linear autoencoding objectives admit infinitely many zero reconstruction error solutions [1, 2], and even sparsity penalties do not eliminate these degeneracies in general. This is precisely why the identifiability conditions introduced in Section 4 are necessary, and why we focus on evaluations on reporting MCC and accuracy of single-concept steering – these are meaningful evaluations that yield high metric values for identifiable representations and low metric values for non-identifiable ones (unlike reconstruction).
>
> _(Response continued due to character limit)_
>
> [1] [Nonlinear Independent Component Analysis: Existence and Uniqueness Results; Hyvärinen and Pajunen](https://www.cs.helsinki.fi/u/ahyvarin/papers/NN99.pdf)
>
> [2] [Neural networks and principal component analysis: Learning from examples without local minima; Baldi and Hornik](https://www.sciencedirect.com/science/article/abs/pii/0893608089900142?via%3Dihub)

---

> > ### Author Response · Authors · 2025-11-21
> > **continued response : 1**
> >
> > > _When presenting a new SAE approach, I'd like to see at least one of two things: (1) the new SAE is better. Then, this needs to be proven empirically, or (2) the new SAE can answer scientific questions we are interested in that normal SAEs could not. This would also need to be shown empirically. I didn't see evidence for either._
> >
> > As we have noted and as the other reviewers also recognize, the goal of this paper is not to present a new SAE approach. The goal is to develop conditions under which concept recovery without concept labels is identifiable, and propose a method that can implement these constraints in practice. We in fact do validate the key claim that we have an identifiable method by showing across multiple real-world (TruthfulQA, Bias-in-Bios, Sycophancy, Refusal) and semi-synthetic (Table 4) language datasets that SSAEs perform well at  (i) identifiable recovery of steering vectors such as these are consistent across several runs of the same model, and the realisation of its potential benefits (ii) recovering correlated concepts, and (iii) improving out-of-distribution (OOD) generalisation– all of which are settings where SAEs are known to struggle. Within this scope, we do provide empirical evidence that SSAEs outperform SAEs on several important axes: out-of-distribution generalisation (Fig 4), ability to steer correlated concepts (Figs 3-4), and effective steering of LLM behaviour through text generations at lower strengths (Fig 2). In other cases, SSAEs are at least competitive with SAEs. In case we are missing something: can you elaborate on the evidence you did not see?
> >
> > > _The author's goal is to steer and they use a supervised dataset with labelled concepts to steer (which SSAE require). However, there are also other methods that can be used to find steering vectors in this setting, for example distributed alignment search. The authors don't compare their SSAE to steering methods that utilize the same dataset._
> >
> > We would like to emphasise that this *is not true*: SSAEs **do not** require datasets with supervised concepts. The labels present in some datasets (e.g., for TruthfulQA or Bias-in-Bios) are used only for evaluation, not for learning the steering vectors themselves. Moreover, for the identifiability claims, we also show evaluations purely without any labels (Tables 1 and 2). By contrast, note that DAS is actually a supervised method: to perform DAS, one requires a fully specified, ground truth model involving known concepts and the outcome of interest. Given that we focus on interpretability methods without supervision, we compare against related methods like SAEs. As noted earlier, the goal of the paper is to provide a theoretically grounded method with provable identifiability properties, not to benchmark empirical steering performance across all existing techniques. In all our experiments, SSAEs are compared to steering methods on the same dataset; can you clarify your concern here: “The authors don't compare their SSAE to steering methods that utilize the same dataset”?
> >
> > We’d like to reiterate that requiring extensive evaluations within the autointerp pipeline sets expectations that extend beyond the intended scope of the paper and its stated contributions. Such demands risk evaluating the work for what it is not, rather than assessing it on its own methodological merits.
> >
> > > _the title is generic and says nothing. What is the one result or idea that you'd like to convey?_
> >
> > The core result we aim  to convey is precisely what is outlined in the title: Identifiability of Concepts from Large Language Model Activations. The paper develops conditions under which concept shift directions can be provably recovered from LLM representations, and introduces an architecture whose objective satisfies these identifiability properties. In case you have specific feedback on the central contribution of our paper being framed differently, please feel free to share your suggestion.
> >
> > _(Response continued in next comment due to character limit)_

---

> > > ### Author Response · Authors · 2025-11-21
> > > **continued response : 2**
> > >
> > > > _imprecision and missing definitions. ll 016 "SAEs are not guaranteed to be identifiable". What does "identifiable" mean? Do you really mean SAEs as a whole or rather SAE features? Why is an absolute guarantee important in the messy world of LLMs?_
> > >
> > > Identifiable representation learning is a decades-old well-established area of machine learning and statistics, and the term has a precise mathematical meaning. We define it formally and provide intuitive explanations at multiple points in the paper (in the Introduction (Section 1); Sections 2, 3, 4). It’d be valuable to know if you are not familiar with the area of machine learning the paper belongs to. Identifiability guarantees are precisely for the “messy world” of unsupervised models, LLMs being one of them. Since empirical behaviour is often unstable, identifiability provides conditions under which recovering concepts is principled and guaranteed rather than accidental. The statement “SAEs are not guaranteed to be identifiable” refers specifically to the fact that neither the features learned by the SAE objective nor the SAE are uniquely recoverable--  can you explain what the difference between the two (the SAE as a whole v/s SAE features) entails in this context?
> > >
> > > > _math is confusing and used heavily throughout the manuscript. However, in most places, it's not needed and just makes the paper very hard to read._
> > >
> > > Can you point to where the math is not needed? The paper makes a theoretical contribution, and math is essential for stating the assumptions, formalising the objective, and proving the identifiability result. Without any motivation, the claims “math is confusing” or “it’s not needed” are impossible to engage with.
> > >
> > > > _important information that would guide understanding like what the dataset is, at least what kind of data it contains, is only given briefly at the end of the paper. Showing data and how it's processed is often easier for the reader's understanding than frontloading unnecessary math._
> > >
> > > This information is provided in detail in Appendix B.1.3.
> > >
> > > > _Different to vanilla SAEs, there's no nonlinearity ...._
> > >
> > > The assumptions made in this comment are incorrect. Sparsity **doesn’t** require ReLU non-linearity. As an example, in sparse coding and dictionary learning– both of which are fields with decades of theoretical foundations– sparse representations are obtained using _linear_ models along with sparsity-inducing objectives or constraints. ReLU is neither necessary nor standard in these settings.
> > >
> > > In fact, sparsity definitely develops through the learning objective being used to update the weights. For example, in the present paper, sparsity arises from the **constrained optimisation objective**, which explicitly enforces a bound on the expected $\ell_1$ norm of the learned SSAE representation. Re: $\ell_0$ metrics, we are happy to report these in the appendix in the revised version of our paper.
> > >
> > > > _Why should the learned representation have size V (ll179)? What do you mean by "learned representation" here? Features? Reconstructions? I'd expect there to be many more concepts than the activation space has dimensions because of superposition._
> > >
> > > The interpretation in this comment is inaccurate because |V| exactly represents the number of differing concepts and identifiability exactly means doing away with superposition to learn a basis where the |V| concepts are represented by separate dimensions. “Learned representations” specifically means the output of the encoder of the SSAE. Your expectation of there being more concepts than the activation space has dimensions is correct and one of the key motivations behind designing SSAEs such that they are identifiable, as we have explained in detail in Sections 2, 3, and 4 both mathematically and intuitively.
> > >
> > > > _Why were activations only taken from the last token in context (ll305)? Unlike e.g. BERT, decoder-only LLMs don't produce a sentence representation at their last token._
> > >
> > > There can exist various other methods for getting activations. Since decoder-only LLMs don’t produce a sentence representation, this is an empirical ablation. We noticed the best performance by considering the last token in our experiments, hence the choice.
> > >
> > > > _ll322 what is "true latent dimension"?_
> > >
> > > True latent dimension would correspond to the number of ground truth latent variables.

---

> ### Author Response · Authors · 2025-11-27
> **Request for Response**
>
> Hi Reviewer whSt, just a gentle reminder to please take a moment to respond to our rebuttal when you have the chance. Your engagement will help us improve our work! Thank you again for your time and feedback.

---

### Official Review · Reviewer_XVbS · 2025-11-03

**Soundness:** 4
**Presentation:** 4
**Contribution:** 3
**Rating:** 6
**Confidence:** 3

**Summary:**

The paper is motivated by recent negative results on using SAEs for LLM steering. The authors hypothesize that this could be due to the identifiability failure of the SAE method. That is, if we assume that model activations $z$ are obtained by a linear map $Ac$ mapping latent concept vectors $c$ in a high-dimensional space to activations $z$ in a lower-dimensional space, it can be provably impossible to recover $c$ by just knowing $z$ (just by linear algebra, $A$ must have a kernel, which opens the door to non-identifiability in principle).

The paper proposes a new training variant for sparse autoencoders (the architecture is almost exactly the usual SAE architecture), which focuses on autoencoding *differences* between pair of activations instead of the individual activations themselves. This allows authors to sidestep the above identifiability problem: when we use differences, we have $z-z' = A(c-c')$, and if we assume that $c,c'$ can only differ in a small, limited set of coordinates $V$, we effectively only care about the part of $A$ applied to the coordinates $V$, $A_V$. $A_V$ may well be invertible.

Relying on some prior work, the authors prove mathematically that with this formulation and assumptions along these lines, their SAE training variant will learn the original concept difference vector $c-c'$ up to coordinate-wise rescaling and coordinate permutation. Importantly, this allows the computation of steering vectors that precisely correspond to a single concept being different between two representations (though it doesn't tell us what the concept encoded by this vector means).

They carry out experiments in three setups:
- using synthetic datasets, compare the correlation between SAE decoders learned by their method from random seeds, and compare this to the analogous quantity for other SAE architectures and training methods. Results show their method outperforms SAE baselines and related representation learning methods.
- using synthetic datasets, compare (using cosine similarity) steering vectors predicted by their method vs ground truth activations encoding the desired change. Again results show an advantage of the method.
- run on realistic datasets with known concepts, and check the correlation between the concepts vs the SAE directions corresponding to them being active. They also apply these steering vectors for text generation, with mixed results.

**Strengths:**

- The paper proposes an inventive way to change the SAE training method that can plausibly improve the identifiability of the concepts found by the SAE by making its task "easier" in a sense. Improvements to SAEs are an interesting and important area of study today, as it becomes clear that SAEs are a useful tool for exploration/model-diffing/data attribution, but still come with a lot of alarming disadvantages in other areas.
- The method is solidly backed up by conceptual and theoretical arguments
- The writing is very clear. Experiments align well with the theoretical story and do not overclaim.

**Weaknesses:**

- Not sure if cosine similarity is the best way to measure closeness to a steering vector; ideally, we would have more causal tests similar to your experiments on the bias in bios dataset.
- On a high level, what we have done here is: made some assumptions about how concepts are mapped to activations, and used them to derive an SAE-type algorithm that can provably recover interesting stuff about these concepts. However, I don't see it clearly addressed in the paper why these gymnastics are necessary, given that we could have plausibly had a similarly convincing story with the original SAE algorithm in place.
	- I'm not an expert on sparse recovery, but my understanding is that under certain assumptions on the matrix $A$ above, an ordinary SAE could also provably recover the concept vector. A quick search turns up e.g. Hu et al., _Global Identifiability of $\ell_1$​-based Dictionary Learning via Geometric Analysis_.
	- Your paper says: "Unfortunately, unconstrained autoencoding objectives are non-identifiable (Hyvärinen & Pajunen, 1999), and sparse autoencoding objectives (Cunningham et al., 2023) may not be able to invert embeddings to potentially billions of concepts." - I guess you are hinting at the fact that billions of concepts might be too much for our SAEs, which are typically much smaller than that; and that your trick would allow you to, in principle, decrease the effective number of concepts to whatever you want.
	- However, SAE also come with a sparsity assumption, so the effective number of concepts should be decreased somehow? It would be great to see this question addressed more thoroughly and categorically in the paper!

**Questions:**

- my main question goes back to the discussion on identifiability in ordinary SAEs - what are your reasons to believe your setup is better in some ways than the ordinary SAE setup + sparsity assumption on the concepts?

---

> ### Author Response · Authors · 2025-11-21
> **Response to Reviewer XVbS**
>
> We thank you for your review of our work; we are delighted that you find our method “inventive” and contributing to the important area of study of “improvements to SAEs”, “solidly backed up by conceptual and theoretical arguments”, and experiments aligned with claims. To avoid any potential confusion, we’d like to clarify a few aspects of our experimental setup. All experiments in the paper are on language datasets (ranging from simple phrases/words (Table 4) to real-world datasets such as TruthfulQA, Bias-in-Bios); ; we do not use synthetic data or synthetic ground-truth concept vectors. Our results consistently reflect the advantages expected by using identifiable steering vectors, so we would not describe them as “mixed.” We are happy to discuss more about this. Below, we engage with the questions and nuanced points raised in your review, and we welcome further discussion on these aspects:
>
> > _The paper proposes a new training variant for sparse autoencoders (the architecture is almost exactly the usual SAE architecture),..._
>
> While you correctly note that the proposed SSAE architecture is structurally close to the standard SAE, we would like to clarify a subtle but important distinction. SSAE differs not only in learning from differences between a pair of activations, but also in employing a constrained learning objective, which is an engineering contribution of the paper. This yields substantive practical benefits: significantly better hyperparameter transfer across datasets and LLMs considered as well as stabler training dynamics, both of which are known issues in the training of SAEs.
>
> > _Not sure if cosine similarity is the best way to measure closeness to a steering vector; ideally, we would have more causal tests similar to your experiments on the bias in bios dataset._
>
> We appreciate your point that cosine similarity may not be the optimal metric for assessing the closeness of a learned vector to a desired steering direction. Indeed, developing richer causal or intervention-based evaluations (similar to the ones in the bias-in-bios setting) would be highly valuable. At the same time, this question is somewhat orthogonal to the contributions of our paper. We stress that the key contribution of this paper is theoretical: we provide a novel result on the provable recovery of steering vectors in the absence of supervision.
>
> Our work contributes a result that was not known in the sparse autoencoder literature, and offers new assumptions and results that go beyond the result from [1] on learning concept subspaces. The choice of how best to apply or evaluate such vectors (whether via cosine similarity, causal tests, or other mechanisms) constitutes a separate empirical question, one that likely requires large-scale evaluation across multiple extraction methods (e.g., mean differencing, SAEs, SSAEs) and multiple steering procedures. We view this as an important direction for follow-up work.
>
> > _On a high level,.._
> > - _I'm not an expert on sparse recovery, … Hu et al…._
>
> Thank you for raising this point and for the helpful reference to Hu et al. (“Global Identifiability of $\ell_1$​-based Dictionary Learning via Geometric Analysis”). There is indeed a rich literature on identifiability of sparse coding and dictionary learning, which in part inspires our work to provide theoretical guarantees together with a practical procedure that is suitable for mechanistic-interpretability settings, particularly those involving multi-concept shift data. However, the conditions under which these theorems apply differ in important ways from the setting we study. For instance, Hu et al. require strong geometric conditions on the dictionary $\mathbf{A}$, including incoherence and independence of supports across the dictionary columns (which represent steering vectors). Intuitively, these assumptions rule out correlated columns of $\mathbf{A}$. In the setting of recovering concepts from an LLM, the concepts we recover steering directions for, are often strongly correlated (for e.g., “truthfulness” and “harmlessness”; or the correlated concepts in our _CORR(2,1)_ dataset), precisely a regime where existing SAE methods do not perform well. Thus, the nature of assumptions on the matrix $\mathbf{A}$ is important to consider. Our work offers one potential theoretically grounded route to identifying concepts under assumptions tailored to the LLM setting. While we are not aware of prior identifiability results of this form for LLM activations, alternative assumptions or formulations can yield complementary approaches in the future.
>
> _(Response continued in further comments due to character limit)_
>
> [1] [Learning Interpretable Concepts: Unifying Causal Representation Learning and Foundation Models; Rajendran et al.](https://arxiv.org/abs/2402.09236)

---

> ### Author Response · Authors · 2025-11-21
> **continued response**
>
> Intuitively, it is not surprising that SAEs _might_ identify concepts in favourable regimes, but there are no theoretical guarantees. A guarantee would imply that the method always identifies the intended concepts. In contrast, our aim is to show that, under a generative model of concept shifts, the SSAE autoencoding objective is identifiable in the presence of a large, correlated concept space. Moreover, existing identifiability results from dictionary learning and sparse coding do not necessarily apply here, as their assumptions (such as, incoherence or independence of supports) might not typically be satisfied in correlated concept spaces, which are common when learning concepts from LLM activations. We appreciate your suggestion, and in the revised version of the paper we will add an appendix section providing a clearer comparison with dictionary learning and sparse coding results.
>
> > _On a high level,..._
> > - _…SAE also come with a sparsity assumption…_
>
> Yes, you are correct that sparsity already appears in standard SAEs, which might seem to reduce the “effective” number of concepts. The key difference in our approach is not merely reducing the number of latent units, but _changing the object being modeled_. In Section 3, we explicitly motivate SSAEs as modeling differences between activations rather than the activations themselves. Under our assumptions, concept shifts are sparser than full concept vectors, and, crucially, they enable us to learn concepts relationally, based solely on how activations change across examples in the data. Concretely, if an activation $\mathbf{z} = \mathbf{Ac}$ mixes many latent concepts, then identifying the absolute set of concepts present in $\mathbf{z}$ is ill-posed without ground-truth labels or a reference point. However, when we compare two activations, we mitigate these issues.
>
> For e.g., consider the activation $\mathbf{z}$ stemming from a phrase $\mathbf{x}$, “Je suis étudiant”. Many concepts co-occur simultaneously– language, topic, formality, sentiment, etc– so a standard SAE has no principled way to decide which of these is “the” French concept. This ambiguity arises for two reasons. First, recovering the absolute concept vector $\mathbf{c}$ would require knowing an explicit origin in concept space, against which this specific phrase can be compared. Second, even if such an origin existed, there is no principled way to determine _how_ many concepts are present in this given phrase since multiple concepts co-occur co-occur and their absolute contributions are fundamentally indeterminable from activations alone.
>
> However, by pairing it with some other phrase (even semantically unrelated), such as, “The conference is huge”, we can identify the concepts that varied among the two, one of which would be _French -> English_.  By considering several such pairs of phrases, we can isolate the concepts that vary consistently across all comparisons, as revealed through their activation differences. This also addresses the second issue: we no longer need to know how many concepts are present in a sentence or what their absolute strengths are. Even if a sentence expresses many latent concepts simultaneously, the difference between any two matched phrases highlights only the concepts that change across them, yielding a much sparser and more identifiable signal.
>
>
> > _my main question goes back to the discussion on identifiability in ordinary SAEs - what are your reasons to believe your setup is better in some ways than the ordinary SAE setup + sparsity assumption on the concepts?_
>
> We thank you for your question. In Section 3 of the paper, we discuss in detail the various design choices that make SSAEs identifiable. We do not claim SAEs are unidentifiable, but highlight the issues in their design which make identifiability difficult and fall outside the scope of existing tools from identifiable representation learning. These issues show up in the empirical results we have observed so far, especially low out-of-distribution generalisation performance, and issues with identifying correlated concepts. Our setup addresses these issues by operating on sparse concept shifts, for which we formally prove identifiability and provide a practical procedure for realising it on real-world data. We are happy to discuss this further in case of any remaining questions.

---

> ### Author Response · Authors · 2025-11-27
> **Request for Response**
>
> Dear Reviewer XVbS, just a gentle reminder to please take a moment to respond to our rebuttal when you have the chance. Your engagement will help us improve our work! Thank you again for your time and feedback.

---

### Meta-Review · Area_Chair_VEmM · 2026-01-09

**Summary:**

The paper studies identifiability of concepts from large language model activations. The core research question is well-motivated and brings a (much needed) theoretical lens to the SAE literature. The identifiability question of SAE concepts posed in the paper is excellent, and may help shed light on why SAEs underperform.

The authors argue that standard sparse auto-encoders (SAEs), in their plainest form, are not guaranteed to be identifiable. They address this by proposing an alternative called "Sparse Shift Auto-Encoders" (SSAEs), which consider differences of embeddings. The paper provides theoretical results claiming that the contrastive approach is, in fact, identifiable. The paper received three fairly detailed reviews and one shallow review (the most positive one). Unfortunately, none of the reviewers engaged in the post-rebuttal discussion process, which makes it harder to determine whether scores would have changed post-rebuttal. My assessment deemphasizes reviewer YXnW score (the highest score -- 8), since their very brief review reflects a rushed reading of the paper.

Among the reviews, XVbS is overall positive and appreciates the theoretical grounding, while also questioning some of the “gymnastics” connecting the paper to prior work in dictionary learning (e.g., global identifiability results for L1-based dictionary learning) and noting that comparisons to standard SAE work could be stronger. whSt is the most critical, raising many concerns about presentation, framing, and missing comparisons to the broader SAE literature, and making requests (e.g., dashboards and extensive benchmarks) that the authors characterize as out of scope (I tend to side with the authors here). AEzo leans towards rejection, raising limitations of both the theoretical assumptions as well as SSAE experiments. Though not as emphatic, these issues echo whSt’s concerns. Again, YXnW gave an extremely brief review, with only a few sentences of strengths and weaknesses, and I place less weight on it than the more detailed reviews.

From my own read, I do not fully agree with the authors' rebuttal stance of the paper’s positioning as primarily a strong theoretical contribution. While the framing of “identifiability of SAEs” seems novel, the identifiability conditions and derivations themselves appear routine: the linear identifiability proposition is elementary, and the permutation identifiability result relies heavily on prior arguments (in particular, Lachapelle et al., 2023) rather than new techniques specific to SSAEs. This creates a tension in positioning the main contributions of the paper: if the core contribution is identifiability theory for SAEs, the results may be too routine to stand on their own; if the core contribution is SSAEs as an identifiable alternative to SAEs, then the comparisons to SAE theory and practice need to be more precise with additional experiments (though perhaps not as the rather unrealistic bar posed by whSt's negative assessment).

**Reviewer Concerns:**

Concerns that the rebuttal addressed (partially): The authors took a firm stance with whSt, emphasizing that requests such as future dashboards and broader benchmark-style evaluations are out of scope for a paper whose primary goal is to propose a provably identifiable SAE variant. I tend to side with the authors on the scope point, in the sense that not all requests for additional tooling or dashboarding should be treated as required for this contribution. The rebuttal also provided clarifications, but it did not introduce substantial new experiments.

Concerns that remain outstanding: Several substantive concerns raised by whSt and AEzo are, in my view, unresolved, largely because they depend on either additional empirical validation or review of an updated manuscript with sharper framing and comparison with prior work. These include: (i) stronger and more precise comparisons with the broader SAE literature, (ii) clearer positioning relative to identifiability and dictionary learning results (raised in different ways by XVbS and AEzo), and (iii) addressing the experimental limitations around SSAEs noted by AEzo (and echoed much more forcefully by whSt). Overall, the paper needs to be firmer and clearer about its end-to-end contribution: whether it is primarily an identifiability-theory paper (in which case novelty and depth are the concern) or primarily an SSAE method paper (in which case comparison and validation are the concern).

When jointly considering the reviews, it becomes clear that the paper would be much stronger if the authors sharpen the comparison to prior SAE work and dictionary learning identifiability results, and revise the framing to clearly match what they see as the paper’s true contribution (theory versus new interpretability method) while making the limits of the identifiability assumptions explicit. Together, these concerns highlight that the paper would benefit from an additional round of re-submission and review.

**Reviewer Scores:**

This paper is in a difficult position since **none** of the reviewers engaged in the rebuttal process. Overall, I believe the rejection-leaning scores would likely have remained unchanged. I particularly skeptical that whSt would have made any changes in their assessment given their original detailed yet very negative review. I am also not convinced that AEzo would change their opinion to a very positive assessment of the paper (say, beyond a overall score of 6).

---

### Decision · Program_Chairs · 2026-01-26

Reject